# A Review of ULK1-Mediated Autophagy in Drug Resistance of Cancer

**DOI:** 10.3390/cancers12020352

**Published:** 2020-02-04

**Authors:** Li Liu, Lu Yan, Ning Liao, Wan-Qin Wu, Jun-Ling Shi

**Affiliations:** Key Laboratory for Space Bioscience and Biotechnology, School of Life Sciences, Northwestern Polytechnical University, 127 Youyi West Road, Xi’an 710072, China; liuli3696352@mail.nwpu.edu.cn (L.L.); 1508271153@mail.nwpu.edu.cn (L.Y.); liaoning@mail.nwpu.edu.cn (N.L.); wuwanqin@mail.nwpu.edu.cn (W.-Q.W.)

**Keywords:** ULK1, autophagy, cancer, drug resistance

## Abstract

The difficulty of early diagnosis and the development of drug resistance are two major barriers to the successful treatment of cancer. Autophagy plays a crucial role in several cellular functions, and its dysregulation is associated with both tumorigenesis and drug resistance. Unc-51-like kinase 1 (ULK1) is a serine/threonine kinase that participates in the initiation of autophagy. Many studies have indicated that compounds that directly or indirectly target ULK1 could be used for tumor therapy. However, reports of the therapeutic effects of these compounds have come to conflicting conclusions. In this work, we reviewed recent studies related to the effects of ULK1 on the regulation of autophagy and the development of drug resistance in cancers, with the aim of clarifying the mechanistic underpinnings of this therapeutic target.

## 1. Introduction

Cancer is one of the leading causes of death in humans [1]. Although significant advances have been made in cancer research, the successful treatment of cancer still faces severe challenges [2]. Chemotherapy is a common form of cancer treatment [3] which is used as monotherapy or in combination with radiation therapy to treat cancers [4]. Chemotherapy drugs, whether they are commonly used cytotoxic drugs or small molecule targeted drugs, can encounter therapeutic barriers due to the development of drug resistance in tumor cells which is a common cause of tumor recurrence and metastasis [5]. Drugs targeting the pathways related to cancer development can improve the prognosis if used in the early treatment of some cancers. However, these targeted treatments can cause drug resistance in tumor cells resulting in the failure of therapies [6,7,8,9]. Tumor resistance can be divided into two categories, i.e., inherent and acquired. Inherent resistance, on the one hand, is defined as tumor drug resistance that occurs even without any drug treatment and is typically caused by genetic mutations. Acquired drug resistance, on the other hand, develops during cancer treatment as an adaptation to selective pressure from the therapeutics. These processes are related to the high expression of therapeutic targets and the activation of compensatory signaling pathways [10,11,12]. 

Autophagy is a system of self-degradation that is a prevalent element of drug resistance in tumors [13]. It can be either beneficial or harmful to the occurrence of drug resistance in tumor cells, either protecting tumor cells from the effects of chemotherapy drugs, or killing multidrug resistant cells [13,14]. Patients with poor prognoses typically have higher measured levels of autophagy relative to patients with good prognoses, suggesting that autophagy can lead to the development of MDR [15]. The effect of autophagy on tumor cells varies according to the tumor type and the stage of cancer development [16]. In the precancerous stage, autophagy eliminates obsolete cellular constituents such as misfolded proteins or damaged organelles from cells [17]. The inhibition of autophagy at this stage leads to the increase of intracellular reactive oxygen species, genomic dysfunction, and the accumulation of P62 protein. These changes collectively result in an increase of ER pressure and DNA damage, and thus promote the formation of tumors [18]. At this stage, autophagy is a tumor-suppressing factor [19]. However, under conditions of starvation or oxidative stress, autophagy can also provide nutrients and energy to established metastatic tumors. In this way, autophagy can paradoxically act as a cancer-promoting factor [20]. Tumor cells can reuse proteins and damaged organelles through autophagy, and therefore survive despite drug treatment. Indeed, here, inhibiting autophagy can promote the death of tumor cells [21]. 

Unc-51-like kinase 1 (ULK1) is a cytoplasmic kinase that is important to the process of autophagy. It is a homologue of ATG1 gene in mammals, with a total similarity of 29% [22,23]. It has been shown that ULK1 can either promote or inhibit tumor growth through protein–protein interactions and post-translational modification-mediated autophagy in nutrient-deficient environments [24]. Thus, both the positive and negative regulation of ULK can situationally protect tumor cells from excessive autophagy [25]. ULK1 forms a protein complex together with the autophagy proteins mATG13, FIP200, and ATG101 to regulate the initiation of autophagy [26]. The induction of autophagy requires the activation of this ULK complex, and the ULK complex is directly regulated by mammalian target of rapamycin (mTOR) and AMP-activated protein kinase (AMPK) [27]. The observation of ULK1-mediated inhibition of the early autophagosome indicates that ULK1 not only participates in the initiation of autophagy, but also regulates the maturation of autophagosomes [28]. In this review, we summarized the biological function of ULK1 in tumors with respect to its potential as a target for tumor therapy and its impact on the occurrence of drug resistance by mediating autophagy in tumor cells.

## 2. Structure of ULK1 and Its Biological Functions in Tumors

The mammalian genome contains a total of five ATG1 homologues, ULK1, ULK2, ULK3, ULK4, and STK36 [29]. Among them, only ULK1 and ULK2 exhibit extensive sequence similarity over the entire protein length [30], whereas ULK3, ULK4, and STK36 only have sequence similarity to the Atg1 gene in the kinase structure domain [31]. The conserved domains of ULK1 in different model organisms have been confirmed (Table 1). Sequence analysis of ULK1 and ULK2 indicates that their kinase domains have 75% sequence identity [32,33]. All of these proteins have an N-terminal serine/threonine protein kinase domain (KD) and a C-terminal interacting domain (CTD). Between the kinase domain and CTD, there is a serine/proline-rich region that is the site of many post-translational modifications. However, the catalytic domain of ULK2 is a dimeric assembly that is different from the typical monomer of ULK1 [34,35]. The KD domain is responsible for the catalysis of kinase activity [36] and the interaction between ULK1 and LC3. The CTD domain contains two microtubules and a MIT domain structure that is a scaffold to enable the interaction of ULK1 with ATG13 and FIP200 [37,38]. ULK1 has a standard kinase fold that has several unique features, including a large loop between the N- and C-terminal lobes [33,39].

Studies have shown that knockdown of ULK inhibits autophagosome formation. Mice with defects in ULK1 and ULK2 die within 24 h after birth [41]. When ULK1 is expressed in mammalian cells, ULK2 is not necessary for autophagy. However, when ULK1 expression is inhibited, ULK2 can compensate for the function of ULK1 in regulating autophagy [42,43]. Recent evidence has revealed that ULK1 deficiency in mice does not affect survival or fertility. The cells still exhibit autophagy but with a delay in autophagic mitochondrial clearance in reticulocytes during erythrocyte development [44]. Knock out of ULK1 and ULK2 in mouse embryonic fibroblasts could destroy the autophagy induced by amino acid or glucose deficiency [45]. However, in varying cell environments, the roles of different ULK subtypes are diverse. It was found that siRNA-mediated silencing of ULK1 in HEK293 cells can completely inhibit autophagy [46]. It has also been reported that ULK1 can degrade rapidly in mouse embryonic stem cells when FIP200 is knocked out. In this case, the stability of ATG13 and ATG101 was also affected [46]. It has been shown that ATG101 in mammals has no homology with other ATG proteins and is a vital molecule participating in autophagy [32,47]. ATG101 regulates the phosphorylation of Atg13 by directly binding to Atg13 [48] and forming a stable complex with ULK1-Atg13-FIP200 [49].

The ULK complex in the phagophore membrane promotes the recruitment and activation of VPS34 in phosphatidylinositol-3 (PI3)-kinase complex I and forms PI3P in the phagophore membrane, a process crucial to the formation of the autophagosome [50,51]. In the state of amino acid deficiency, ULK1/2 are activated and transported to the ER to promote autophagosome nucleation [52,53]. Under conditions of severe hypoxia and ER stress, ATF4 (activating transcription factor 4) directly regulates transcription and expression of ULK1 [54,55]. The binding of ATG8 and ULK1 plays a vital role in the formation of the phagophore and autophagosome [56]. 

The mammalian target of rapamycin protein plays a fundamental role in sensing extracellular nutrition, regulating the rate of protein synthesis and dictating cell growth [57]. Under normal nutritional conditions, mTOR directly binds to ULK1 via the RAPTOR subunit to form a complex which inhibits the occurrence of autophagy by the phosphorylation of ULK1 (at serine 638 and 758) and ATG13 (at Ser258) [58]. When nutrients are sparse, mTOR activity is inhibited, the dephosphorylation of Atg13 is promoted, and the mTOR and ULK1 complex is separated. After separating from mTOR, ULK1 is activated and forms a complex with Atg13 and FIP200/Atg17, and thus promotes autophagy (Figure 1) [59,60,61]. In addition, in the absence of energy, AMPK is activated, and mTOR-induced autophagy is negatively regulated by the phosphorylation of ULK1 at s555 [62,63]. This complex process is summarized in Figure 1.

### 2.1. The Canonical Role of ULK1

The canonical functions of ULK are related to autophagy [64]. Autophagy is a highly evolutionarily conserved lysosomal degradation pathway in which intracellular misfolded proteins and damaged organelles are degraded [65]. Studies have suggested that maintenance of the correct level of autophagic activity is crucial to maintaining muscle integrity, with either reduced or excessive levels leading to specific myopathies. The ubiquitin ligase TRIM32 stimulates ULK1 activity via unanchored K63-linked polyubiquitin to regulate muscle autophagy in response to atrophic stimuli [66,67,68]. Autophagy includes the following four stages: autophagy initiation, autophagosome formation, autophagosome-lysosome fusion, and the degradation of autophagic substrates [69]. It has been shown that tumor of the central nervous system with the BRAFV600E mutation are autophagy-dependent, and late stage inhibition of autophagy improves the response to targeted BRAF inhibitors (BRAFi) in sensitive and resistant cells [70]. Autophagy is a rapid response to a variety of stress conditions that is carried out via the concerted action of evolutionarily conserved proteins, most of which are known as autophagy-related (ATG) proteins that function at different steps [71,72] (Table 2). Under normal physiological conditions, autophagy helps maintain cell homeostasis mainly through the control of proteins and organelles, which is called basic autophagy [73]. However, under stressful conditions, such as hypoxia or deficiencies of ATP or amino acids, intracellular autophagy is induced [49,74,75]. Upon sensing changes in cellular energy, the ULK complex acts as a scaffold to initiate the phagophore assembly site (PAS) and the recruitment of downstream ATG proteins that play an important role upstream of pathways of autophagy [76,77].

It has been reported that the ULK complex can be recruited to two kinds of distinct membranes, the ER membrane and ATG9-positive autophagosome precursors [78]. After interaction between ULK complex, the PIS-enriched ER subdomain and ATG9A vesicles, the process of autophagy is initiated [27,79]. The mammalian target of rapamycin complex (mTORC1) and the adenylate-activated protein kinase (AMPK) directly participate in the regulation of ULK1 activity [49]. In both a mouse model and in primary hepatocytes, loss of AMPK or ULK1 resulted in aberrant accumulation of mitochondria and the autophagy adaptor p62. However, a study of ULK1-deficient cells found that the addition of a plasmid encoding a mutant ULK1 that could not be phosphorylated by AMPK revealed that phosphorylation is required for mitochondrial homeostasis and cell survival during starvation [79]. Under nutrient-rich conditions, autophagy is inhibited because AMPK is inhibited and the activity of ULK1 is inhibited by mTORC1 via the phosphorylation of ULK1 at Ser757 and Ser258 of ATG13 [80]. Under conditions when nutrient supply is limited, however, MTORC1 separates from the ULK complex and induces autophagy by ULK1 autophosphorylation and transphosphorylation of ATG13 and RB1CC1 [47]. Meanwhile, ULK1 causes phosphorylation of ATG9 at Tyr8 and Ser14 to promote the redistribution of ATG9 from the plasma membrane and Golgi region to the sites of autophagosome assembly [81]. ULK1 binds to ATG8 by partially recruiting its LIR motif to autophagosomes. Mutation of the LIR motif of ULK1 can cause the accumulation of early autophagic structures [82,83].

Once the active ULK complex is recruited to the initiation site, autophagy proceeds to the next stage, i.e., the formation of autophagosomes. It has been shown that the Beclin1-VPS34-Atg14L complex is critical to the formation of autophagosomes [84]. Activated ULK1 phosphorylates Beclin1 at Ser15 and ATG14 at Ser29. Beclin1 positively regulates Vps34 to increase the activity of the Beclin1-Vps34-Atg14L complex [85]. Matured autophagosomes are degraded by autolysosomes in a manner dependent on small GTPase RAB7, CCZ1-MON1, HOPS complex, syntaxin-17, and other SNARE proteins [86,87]. In mammalian cells, ULK1 binds to ATG8 at the interacting motif/LC3-interacting region and, then, is transferred to the lysosome for degradation [88,89,90]. The degradation of ULK1 not only involves lysosomes and proteasomes but also the transcription of inhibitory proteins. The inhibition of protein synthesis leads to a rapid decrease in ULK1 and limits ULK1-dependent autophagic degradation [91,92]. All of these functions are summarized in Table 2.

### 2.2. The Noncanonical Role of ULK1

In recent years, multiple reports have shown that ULK1 not only plays a canonical role in the occurrence and progress of autophagy, but also mediates or participates in many other important physiological activities [93]. For example, ULK1 plays a role in protein transportation, the promotion of apoptosis, and the regulation of the innate immune response [94]. Kemp et al. reported that UV radiation and chemical carcinogens enhance STING-dependent IRF3 activation, which exacerbates the symptoms of autoimmunity. Of note, ULK1 negatively regulates STING [95]. The natural immune response that is induced by ultraviolet light can be enhanced by downregulating the expression of ULK1 in keratinocytes [95,96]. The lack of ULK1 tends to result in impaired RBC macrocytosis and mitochondrial clearance [88], which leads to a shortened life of RBCs [97]. It has been reported that ULK1 has an important function in the activation of the spindle assembly checkpoint (SAC), which ensures the fidelity of chromosome segregation during mitosis. Of note, deletion of ULK1 increases chromosome instability and cytotoxicity following treatment with paclitaxel, resulting in significant impairment of cancer cell growth [98]. 

ULK1 is also reported to be important for neuronal development through mediating vesicular transport in axons [97,99]. In the absence of exogenous pressure, ULK/Atg1 regulates the transportation of specific macromolecules in the endoplasmic reticulum (ER)-Golgi apparatus to maintain cell homeostasis in Caenorhabditis elegans and mammalian cells [50,100,101]. In conditioned knockout mice, ULK1/2 results in neuronal degeneration, but there is no accumulation of P62, ubiquitinated protein, or abnormal membranous structures in nerve cells [102]. In mice, knockout of ULK1 improves the cognitive ability after traumatic brain injury (TBI) treatment and slows the histological changes in the hippocampus. Indeed, loss of ULK1 reduces hippocampus inflammation and apoptosis in TBI mice [103]. The mouse ULK1/2 proteins are localized to vesicular structures in the growth cones of mouse spinal sensory neurons. The Ulk1/2 proteins are necessary for efficient endocytosis of nerve growth factor (NGF) and can suppress excessive filopodia extension and axon branching in sensory neurons during NGF-induced axon outgrowth. Knockdown of ULK1 or ULK2 by RNAi results in impaired endocytosis of excessive axon arborization and severely stunted axon elongation [104,105,106]. 

According to the recent research reports, in the absence of energy metabolism, ULK1 can phosphorylate Rab9 at Ser179 to protect the heart from myocardial ischemia [107]. High-grade serous ovarian cancers (HGSOCs) grow in suspension and potentially escape from anoikis. However, knockdown of ULK1 significantly inhibits suspension growth of HGSOC cells, and analysis of the metabolic profile confirmed an important role of fatty acid metabolism in survival in suspension [108,109,110,111]. ULK1 regulates lipid metabolism to prevent heart dysfunction caused by obesity [112]. The differentiation of human bone marrow and the pro-osteogenic effect of galectin 3 are affected by the knockout of ULK1 [94]. ULK1 and EGFR levels in patients with normoalbuminuria are significantly higher than in microalbuminuria and macroalbuminuria [113]. ULK1 plays an important role in protecting hosts from infection by pathogens; inhibition of ULK1 expression prevents the death of host cells infected by Staphylococcus aureus [114].

## 3. ULK1 Mediates Signaling Pathways in Cancer Treatment

Autophagy is usually understood as a mechanism to protect normal cells from adverse conditions [115]. However, it is also known to inhibit the growth of tumors. Some studies have suggested that the role of autophagy can vary across stages of tumor development [116,117]. For example, in the early stage of tumor development, autophagy limits the proliferation of tumor cells by direct killing. However, after tumors have already formed, the response to stress can actually increase the survival, invasion, and metastasis of tumor cells [118]. In addition, autophagy has been found to influence the occurrence of tumors in cell- or tissue-specific ways. Therefore, autophagy plays a dual role in the development of tumors [72,119]. Compared with other cell functions, autophagy is relatively selectively druggable. During autophagy, only a few of enzymes are currently targeted by drugs, although we know of more than 36 autophagy-related proteins [120]. ULK1 plays a core role in the activation of the autophagy pathway. Therefore, it could be a feasible drug target. As a promoter of autophagy, ULK1 also plays different roles according to the types and stages of tumors. For example, it has been shown that low expression of ULK1 in operable breast cancer tissues is a marker of poor prognosis [121].

### 3.1. Inhibition of ULK1-Mediated Autophagy for Cancer Treatment

Autophagy provides building blocks and energy to tumor cells in response to metabolic stress and chemotherapeutic drug damage, thereby promoting the survival and development of tumor cells [122]. A growing number of studies have shown that the inhibition of autophagy an effective method for tumor therapy [123,124]. Many proteins are involved in this complex process. Thus, targeting these proteins can be one method to inhibit autophagy [125]. A strategy based around the inhibition of kinases has been successfully used in clinical application, making ULK1 an attractive candidate for inhibiting autophagy [126].

The crystal structure of ULK1 has been determined [127]. Petherick et al. reported two compounds, MRT67307 and MRT68921, that were effective in inhibiting ULK1 (IC_50_ values of 45.0 and 2.9 nM, respectively) and ULK2 activity IC_50_ (IC_50_ values of 38.0 and 1.1 nM, respectively) in an in vitro kinase assay and that also prevented autophagy in mouse embryonic fibroblasts (MEFs) cells [128]. In studies of a drug-resistant ULK1 mutant cell line, it was found that the compound inhibits the autophagy of tumor cells by specifically inhibiting the activity of ULK1 [128]. To increase the selectivity for ULK1, a novel inhibitor scaffold of ULK1 was explored, leading to the finding of several compounds that are potent inhibitors of ULK1 and ULK2 with good selectivity [127]. SBI-0206965 is a FAK inhibitor that has been demonstrated to be a highly selective inhibitor of ULK1 and ULK2 that acts by suppressing ULK1-mediated phosphorylation of Vps34 in tumor cells [129,130]. In clinical therapy, SBI-0206965 synergizes with mTOR inhibitors to kill cancer cells [131]. 

Martin et al. found two small-molecule compounds, ULK-100 and ULK-101, that inhibit the autophagy of tumor cells through efficiently and selectively inhibiting ULK1, thus making tumor cells more sensitive to nutrient deficiency. In in vitro tests, the EC_50_ of ULK-100 and ULK-101 were 83 nM and 390 nM, respectively. ULK-101 blocks LC3-II accumulation, and thus inhibits autophagy by stopping the formation of the early autophagy body. The inhibition of ULK-101 causes KARK-mutant tumor cells to be more sensitive as compared with wild cell in conditions of nutrient deficiency. However, the authors stated that further validation is still required in vivo and that the targeted therapeutic mechanisms of ULK1 are not suitable for all genetic or environmental contexts [132]. Liu J. et al. found that NVP-BEZ235, a dual PI3K/mTOR inhibitor, could induce autophagy through AMPK/ULK1 pathway in colon cancer cells. Autophagy was blocked by knocking down AMPK or ULK1 inhibiting cell proliferation and further promoting NVP-BEZ235 induced apoptosis [133]. Arsenic trioxide (ATO) and thalidomide (THAL) are used in the treatment of many types of hematologic malignancies. ATO prevents the proliferation of cells and induces apoptosis in some cancer cells. Moreover, THAL has immunomodulatory and antiangiogenic effects in malignant cells. Combined treatment with ATO and THAL inhibits proliferation and invasion of AML cells by downregulating ULK1 and BECLIN1 and upregulating PTEN and IL-6, and this effect was stronger than the effects of either ATO or THAL alone [134]. The IC_50_ for each inhibitor of ULK1 and ULK2 is listed in Table 3.

### 3.2. Activation of ULK1-Mediated Autophagy for Cancer Treatment

During the past decades, extensive research efforts have greatly improved our knowledge about autophagy. This process is now known to be widely implicated in pathophysiological processes including cancer, metabolic, and neuro degenerative disorders, making it an attractive target for drug discovery [138]. Recent publications have suggested that ULK1 is underexpressed in some tumor tissues, such as breast cancer [139]. This indicates that the activation of ULK1 to inhibit tumor growth has potential to be used as an effective treatment method for some tumors. Through in silico high-territorial screening and chemical synthesis, Ouyang et al. obtained a small molecule activator of ULK1, LYN1604, which regulated ULK1 and interacted with protein ATF3, RAD21, and CASP3/caspase3 to induce autophagy-associated cell death in triple-negative breast cancer cells [139]. LYN-1604 has been shown to be a potent ULK1 agonist with an EC_50_ of 18.94 nM in an in vitro kinase assay [136]. It has been suggested that fluoxetine could have significant therapeutic effects on triple negative breast cancer. It can induce autophagic cell death by activating the AMPK-mTOR-ULK axis in cells. In addition, fluoxetine can also induce apoptosis in TNBC cells by upregulating caspase3/8 and PARP expression [140]. 

A growing number of cell- and animal-based assays have shown that structurally different compounds from various natural products have the same chemotherapeutic effects on tumors [141]. It has been reported that an aqueous extract of clove can inhibit the growth of tumors by activating the AMPK/ULK1 pathway, thus modulating autophagy. AEC inhibits the growth of pancreatic ASPC-1 and colon HT-29 cancer cells in vitro at an IC_50_ of 150 μg/mL in colon HT-29 cells [137]. Although many natural products have health benefits and therapeutic effects on tumors, their poor water solubility, low bioavailability, and rapid elimination limit their practical application as drugs [142]. However, several combinations of natural products and nanotechnology have been tested to create natural product-based nanoformulations. The formed nanoparticles have better affinity, fewer side effects, and can target tumor cells to increase the therapeutic efficacy and improve prognoses for patients [143]. 

The roles of ULK1 in inducing autophagic cell death or cytoprotective autophagy in cancer are summarized in Figure 2.

## 4. ULK1 and Cancer Drug Resistance

With the rapid development of targeted drugs and cytotoxic drugs, most cancers are treatable, and some tumors can often even be completely controlled. However, some metastatic tumors are still incurable [144]. Drug therapy of tumors (including chemotherapy and targeted drugs) is currently the most widely used form of cancer treatment. However, unfortunately, it is usually only effective at the beginning of treatment. With the prolongation of the treatment cycle, the development of resistance to these drugs in tumor cells is often inevitable [145,146]. Multidrug resistance (MDR) is a leading cause of morbidity and mortality in cancer [147]. Drug resistance in tumor cells is caused by a variety of factors including genetic differences in tumor cells and microenvironment. It has been shown the tumor cells from the same tissue source are differently cytogenetic. The microenvironment around tumor cells is essential for the development of tumor resistance [148]. It has been reported that the extracellular milieu around tumors plays an important role in tumor development, metastasis, and the regulation of resistance in various treatments [149]. Although the drug resistance of tumor cells is inherent, acquired resistance is becoming more and more common [150,151,152].

### 4.1. The Pivotal Role of ULK in the Development of Drug Resistance in Human Cancers

The purpose of precision medicine is to develop a personalized treatment in order to address the diversity of cancers. The targeted treatment of tumors is one such strategy and is one of the most effective cancer treatment methods [153]. However, tumor tissue is a multicell system that continues to evolve. Although some drugs and treatments have clinical significance, in the process of continuous treatment, some remaining cells continue to proliferate and, then, acquire drug resistance [154]. Autophagy which is essential to the efficacy of anticancer drugs, as well as drug resistance can have a prosurvival role in response to metabolic and therapeutic stresses [155]. Xia et al. demonstrated that the mitotic kinase NEK2 is involved in the development of MDR by regulating autophagy in multiple myeloma (MM). Autophagy is activated by the assembly of a complex of NEK2/USP7/Beclin-1 in MM cells. Indeed, treatment with a combination of the autophagy inhibitor chloroquine (CQ) and the chemotherapeutic bortezomib (BTZ) significantly prevents NEK2-induced drug resistance in MM cells [156]. The efficacy of fluorouracil (FU) in the treatment of colorectal cancer (CRC) is greatly limited by drug resistance. In this context, autophagy serves as a prosurvival mechanism during chemotherapy challenge [5]. The inhibition of autophagy has been shown to counteract the resistance to doxorubicin in breast cancer MCF-7 cells [157]. ULK1 is crucial in the regulation of autophagy in cancer cells; the phosphorylation of ULK1 alters the activity of autophagy [158]. TOPK (T-LAK cell-originated protein kinase) directly binds with and phosphorylates ULK1 at Ser469, Ser495, and Ser533. The phosphorylation of ULK1 decreases the activity and stability of ULK1, thus, inhibiting autophagy and promoting resistance against temozolomide in glioma cells [159]. The treatment of tumors by targeting ULK, however, has presented many challenges in the intrinsic and acquired drug resistance in cancer treatments [155] (Figure 3).

### 4.2. Drug Resistance-Associated Mutations

Mutations in drug-targeted proteins lead to changes in protein conformation and insensitivity to anticancer drugs of tumor cells [160]. It has been reported that tyrosine kinase inhibitors are effective to treat chronic myeloid leukemia (CML) by targeting the BCR-ABL protein in CML cells [161,162]. However, the development of drug resistance in patients during treatment is mainly caused by a mutation in the BCR-ABL protein [163,164]. IIB02, the inhibitor of Hsp90, which has an inhibitory effect on a variety of tumor cells, could solve this problem by causing significant cytotoxicity to BCR-ABL cells regardless of their sensitivity to tyrosine kinase inhibitors [165,166,167]. This is due to the fact that BIIB021 can significantly induce autophagic cell death by the downregulation of the AKT-mTOR pathway and the activation of ULK1. Additionally, BIIB021 induces apoptosis by releasing of cytochrome c, which promotes consequential activation of caspase-9, -3, and PARP [167,168]. It has also been found that the mutation of BRAF protein in patients with colorectal cancer has an adverse prognosis, causing mortality rates to increase by 70% as compared with normal BRAF backgrounds [169,170]. Although BRAF inhibitors have been approved by the Food and Drug Administration, their therapeutic benefits for cancer patients with BRAF protein mutations are still not obvious [171]. The main reason why BRAFV600E-treated colorectal cancer (CRC) develops drug resistance to selective BRAF inhibitors is that BRAF inhibitors can induce autophagy through the AMPK-ULK1 pathway [172]. Inhibiting autophagy by pharmacological methods or siRNA, however, causes BRAFV600E CRC cells to be more sensitive to BRAF inhibitors to induce cell death and overcome the drug resistance-related mutations in tumor cells [167,173].

Findings from the literature suggest that about 50% of colon cancers have mutations in KRAS, a proto-oncogene that increases resistance to drugs [174]. Mutated KRAS regulates intracellular energy metabolism. When two mitochondrial-targeted compound 3-carboxyl proxyl nitroxide (Mito-CP) and mito-metformin were used to treat KRAS mutated cells, they not only decreased the ATP levels in the tumor cells but also inhibited ATP-related oxygen consumption in cells. However, no obvious effect was found in normal intestinal epithelial cells. Mitochondria-targeted drugs have the capability to activate an AMPK-ULK1 signaling cascade could overcome the resistance of mutant cells to drugs [175,176].

### 4.3. Immune Factors in the Tumor Microenvironment and Tumor Drug Resistance

The tumor microenvironment is the surrounding space composed of immune cells, stroma, and vasculature. The tumor microenvironment mediates drug resistance via several mechanisms, such as preventing immune clearance of tumor cells, hindering drug absorption, and stimulating paracrine growth factors to signal cancer cell growth [177,178]. Recently, a number of studies have shown that the tumor microenvironment could contribute to the regulation of tumor development, metastasis, and drug resistance against various therapeutic methods [179]. Until now, although significant advances have been made in chemotherapy and radiotherapy of tumors, the development of drug resistance during treatment reduces the effectiveness of drugs [180,181]. The tumor microenvironment is a dynamic network of tumor cells and extracellular matrix, usually lacking oxygen and nutrients and presenting a low pH [182]. Most tumor cells adapt to this harsh environment and make use of the limited resources in the environment to grow. Tumor cells can evade immune surveillance by inhibiting immune cell activity [183,184]. It has been reported that the natural product, rocaglamide (RocA) could enhance NK cell-mediated lethality, inhibit the growth of tumor cells, and shrink tumors in in vitro and in vivo tests. RocA not only improves the level of NK cell-derived GZMB and enhances the killing power of NK cells, but also targets ULK1, specifically inhibiting the translation of the ULK1 protein, and therefore inhibiting the progress of autophagy. The inhibition of autophagy increases the sensitivity of non-small cell lung cancer cells to NK cells. However, after inhibiting the activity of NK cells in mice with normal immune function, RocA’s inhibitory effect on tumor inoculation in mice was significantly weakened, indicating that the lack of NK cells could lead to the resistance of tumors to RocA [185,186].

### 4.4. MicroRNA and LncRNA Involved in Drug Resistance

The heterogeneity of tumor cell genes and the differences in transcription process limit the efficiency of many therapeutic methods [187]. MicroRNAs are a class of extremely conserved single-stranded noncoding RNAs of about 19 to 25 nucleotides that negatively regulate the expression of different genes at the transcriptional or post-transcriptional level [188,189,190]. MicroRNAs, as tumor suppressor genes or oncogenes, regulate the occurrence and development of tumors [191]. Increasing amounts of data have indicated that microRNAs play an important role in the development of drug resistance in tumors [192]. Three compounds, doxorubicin (DOX), tamoxifen, and cisplatin have been shown to be effective in the treatment of breast cancer [192]. However, due to inducing drug resistance in tumor cells, their wide application has been limited. They could increase autophagy in cells and induce protective autophagy, which ensures the survival of tumor cells in an adverse environment [193]. Mir-489 has been reported as a tumor suppressor in the treatment of breast cancer cells affecting the activity of multiple genes during autophagy [194,195]. When mir-489 was used to treat doxorubicin-, tamoxifen-, or cisplatin-resistant cell lines, it showed significant inhibition of cell proliferation [196]. It was found that mir-489 could directly target ULK1 and LAPTM4B genes, negatively regulate the expression of ULK1 and LAPTM4B, hinder the fusion of autophagosomes and lysosomes, inhibit autophagy, and thus overcome tumors’ drug resistance [197,198,199].

Long noncoding RNA (LncRNA) is a kind of RNA with no protein-coding capacity that is also over 200 nucleotides in length [200,201]. LncRNA affects protein regulation at transcriptional, post-transcriptional, and epigenetic levels [202]. A growing body of evidence suggests that abnormal expression of LncRNA contributes to tumor formation, metastasis, and drug resistance [203]. It has been reported in the literature that LncRNA plays an important role in the drug resistance of non-small cell lung cancer (NSCLC). The lncRNA small nucleolar RNA host gene 6 (SNHG6) enhances colorectal cancer cell resistance to 5-fluorouracil (5-FU), promotes autophagy, and inhibits 5-FU-induced apoptosis in vivo. Indeed, SNHG6 can promote chemoresistance through ULK1-induced autophagy by reducing the availability of miR-26a-5p in CRC cells. Cancer cells that had knocked down SNHG6 (by SNHG6-specific shRNAs), overexpression of miR-26a-5p, or that were treated with 3-MA were more sensitive to 5-FU [204,205]. Crizotinib showed a therapeutic effect on patients with NSCLC, but after several cycles of treatment, the patient’s resistance to crizotinib increased, leading to treatment failure [203]. HOTAIR (HOX transcript antisense intergenic RNA) is often highly expressed in NSCLC and promotes cisplatin resistance in NSCLC. The silencing of HOTAIR reduced the proliferation and induced apoptosis of NSCLC cells (A549). In addition, HOTAIR shRNA transfection inhibited the resistance of A549 cells to crizotinib, inhibited cell survival, and promoted apoptosis as compared with the HOTAIR scramble group. After HOTAIR was silenced, the number of LC3+ puncta and the expression of Beclin1, p-ULK1, and the ratio of LC3 II/I/in crizotinib-treated A549 cells decreased. Further studies have indicated that the main reason for HOTAIR silencing to reduce the resistance of NSCLC cells could be the inhibition of the phosphorylation of ULK1, thus inhibiting the autophagy of crizotinib-resistant cells [158].

### 4.5. Small Molecule Drugs Targeting ULK Reverse Tumor Resistance

As noted herein, ULK1 plays an important role in the initiation of autophagy [206]. The induction of protective autophagy to inhibit apoptosis is one of the reasons for the development of drug resistance in tumor cells during therapy [207]. Some small molecule drugs targeting ULK1 show inhibitory effects on ULK1 expression and the activity of autophagy, and cause tumor cells to be more sensitive to chemotherapeutic drugs [136,208]. It has been reported that overexpression of ULK1 is inversely related to the prognosis of various tumors, such as colon cancer, breast cancer, lung cancer, nasopharyngeal cancer, and esophageal cancer [136]. The knockdown of ULK1 in NSCLC cells induces an increase in apoptosis and makes them more sensitive to cisplatin [209]. SBI0206965, a selective inhibitor of ULK1, can significantly reduce the cell survival of cisplatin-resistant NSCLC cells by decreasing the conversion of LC3 I to LC3 II, upregulating the expression of autophagy substrate P62, and inhibiting the progress of autophagy. A combination of SBI0206965 and cisplatin can block cisplatin-induced autophagy and promote cell death. However, inhibition of autophagy by SBI0206965 was one of the major reasons for overcoming the resistance of cisplatin to NSCLC cells [131,210]. In addition, more and more natural products are being developed to treat tumors. Most of them are multitargeted drugs, and the inhibition of tumors is achieved through multiple simultaneous effects. However, they can sometimes produce more adverse consequences, such as inhibiting the proliferation of healthy tissues and causing structural deformities, systemic toxicity, long-term side effects, drug resistance, and some psychological problems [140,211,212]. The relationship between ULK1 and cancer drug resistance is summarized in Table 4.

## 5. Conclusions and Future Prospective

Drug resistance is a major problem in cancer therapy. The development of drug resistance in tumors counteracts the therapeutic effects of chemotherapeutic compounds, which leads to a more aggressive recurrence of tumors, and worse prognoses of cancer patients. So far, the solutions to tumor resistance have been mainly focused on selecting more sensitive drug targets, genetically modifying the target, changing the drug structure, using drugs in combination, inhibiting prosurvival pathways, etc. With an increasing number of anticancer drugs on the market, the improvement of preclinical models, and the development of technologies such as high-throughput screening, there are more opportunities to understand and overcome tumor resistance. 

Mammalian Unc-51-like kinases 1 and 2 (ULK1 and ULK2) belong to the ULK/Atg1 family of ULK and are a promising therapeutic target for tumors because they are a direct target of energy- and nutrient-sensing kinases. ULK1 also mediates the adverse prognosis and drug resistance of tumors. Both the inhibition and activation of ULK1 have significant effects on tumor therapy. Regulation of ULK1 plays a fundamental role by influencing cell growth and the development of drug resistance in tumor cells. Although the use of ULK inhibitors as monotherapy have provided modest effects, combining them with other anticancer drugs could help overcome the development of drug resistance during tumor treatment.

Although the role of ULK in the progress of autophagy is clear, other functions of ULK beyond this are not well understood. Future studies are still needed to reveal the presence of further mechanisms. In addition, most studies, to date, have mainly focused on the expression of the ULK gene. With the advancing development of precision medicine, it is crucial to identify more potential predictive biomarkers for tumor treatments, such as ULK inhibitors. This would facilitate the effective design of clinical trials that would accelerate the introduction of these compounds to clinical practice in the most efficient and beneficial manner. Hence, cancer patients would benefit from a tailored therapy with various ULK inhibitors alone or in combination with other molecular targeted therapies.

## Figures and Tables

**Figure 1 cancers-12-00352-f001:**
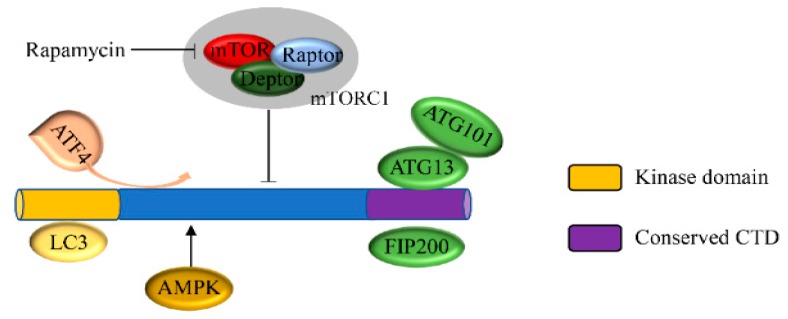
Structure and biological function of ULK1. The domain of ULK1 contains three units including a kinase domain (KD), proline/serine-rich (PS) region, and C-terminal domain (CTD). AMPK and mTORC1 are upstream kinases that regulate ULK1. ATF4 is an activating transcription factor that directly regulates transcription of ULK1.

**Figure 2 cancers-12-00352-f002:**
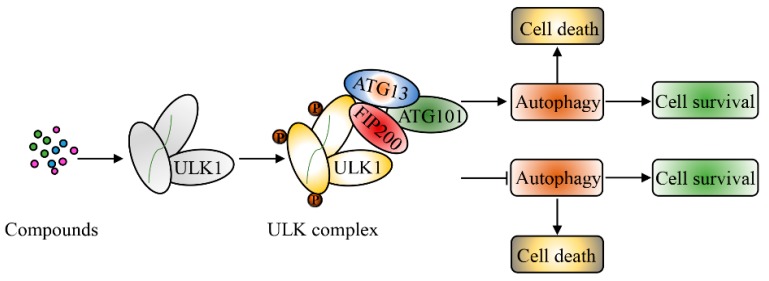
Summary of the relationships between ULK1, autophagy, and cancer.

**Figure 3 cancers-12-00352-f003:**
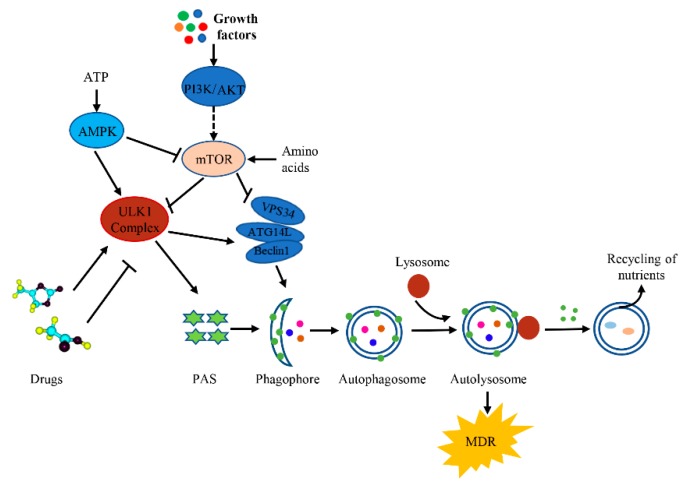
A summary of how ULK1 is involved in the development of drug resistance in cancer.

**Table 1 cancers-12-00352-t001:** Identity of ULK1 in Homo sapiens as compared with its homologues in other model organisms [28,29,30,31,32,33,34,40].

Homologues Name	Model Organisms	Open Reading Frame (ORF)/Amino Acids	N-Terminal Kinase Domain/Residue	Proline/Serine-Rich (PS)/Residue	C-Terminal Domains/Residue
ULK1	*Homo sapiens*	1050	16–278	279–832	833–1050
ULK1	*Mus. musculus*	1051	26–278	279–828	829–1051
ULK2	*Homo sapiens*	1036	9–271	272–811	812–1036
ATG1	*S. cerevisiae*	897	24–325	326–740	741–897
UNC-51	*C. elegans*	856	9–275	276–631	632–856

**Table 2 cancers-12-00352-t002:** Induction factors, important autophagy-related proteins, and their relevance to different stages of autophagy in mammalian cells [73,87,89,91,92,93,94].

Induction Factors	Autophagy Stages	Complex	Core Proteins of the Complex
**Intrinsic cellular stress**: Reactive oxygen species (ROS), damaged organelles, protein aggregates, infected pathogens, AMP/ATP, endoplasmic reticulum stress and hypoxia, etc.	Induction of autophagy	mTORC1 complex	AMPK mTOR, RAPTOR, mLST8, PRAS40, DEPTOR, CASTOR1, SLC38A9, GATOR1
	Nucleation and expansion of phagophore, Formation of autophagosomes	ULK1-FIP200-ATG13complex Beclin-1-VPS15-VPS34-ATG14 l complex	ULK1, ULK2, ATG13, ATG101, FIP200 ATG9, VPS34, VPS15, BECN1, ATG14L, AMBRA, ATG5, ATG12, ATG16
**Extrinsic cellular stress:** Nutrients (amino acids, glucose, and oxygen), growth factors and drugs, etc.	Fusion of autophagosome-lysosome fusion	Lipidation complex	WIPI1/2, LC3, ATG3, ATG4, ATG7, ATG12, ATG16L, GABARAP
	Degradation of autophagic substrates		syntaxin-17, SNARE, VAMP8, SNAP29, RAB7, EPG5, HOPS, PLEKHM1

**Table 3 cancers-12-00352-t003:** Small-molecule compounds targeting ULK1/ULK2-mediated autophagy in cancer.

Function	Compound	In Vitro Kinase Assay (nM)		CellularEC50/IC50 (nM)	Cell Type	Reference
		ULK1 IC_50_	ULK2 IC_50_			
ULK1/ULK2 inhibitor	Compound 1	5.3	13	-	-	[127]
	Compound2	67	200	-	-	
	Compound 3	120	360	-	-	
	BX-795	87	310	-	-	
	Compound 6	8	-	-	-	[33]
	MRT67307	45	38	-	MEFs	[128]
	MRT68921	2.9	1.1	-		
	SBI0206965	38	212	EC50 = 2400	U2OS cells	[135]
	ULK100	1.6	2.6	EC50 = 83		
	ULK101	8.3	30	EC50 = 390		
	NVP-BEZ235	-	-	-	Colon cancer cells	[133]
	ATO -THAL	-	-	-	U93, KG-1 cells	[134]
ULK1/ULK2 activator	LYN-1604	18.94	-	IC50 = 1.66 μM	MDA-MB-231	[136]
	AEC	-	-	150 μg/mL	HT-29 cells	[137]

**Table 4 cancers-12-00352-t004:** Molecular events of ULK1-related cancer drug resistance.

Drug Resistance Cancer Cell Type	Event	Effects on Cancer Drug Resistance
Chronic myeloid leukemia (CML) resistance to tyrosine kinase inhibitors	A mutation in the BCR-ABL protein; IIB02 producing significant cytotoxicity to BCR-ABL mutation cells BIIB021 inducing autophagic cell death by the downregulation of AKT-mTOR pathway and the activation of ULK1	Overcoming drug resistance
Colorectal cancer (CRC) resistance to selective BRAF inhibitors	A mutation in BRAF protein; BRAF^V600E^ of colorectal cancer (CRC) develops drug resistance to selective BRAF inhibitors is that BRAF inhibitors induce cytoprotective autophagy through AMPK-ULK1 pathway	Inhibiting autophagy and overcoming drug resistance
Colorectal cancer cell resistance to 5-fluorouracil (5-FU)	SNHG6 enhances chemoresistance through ULK1-induced autophagy via reducing free miR-26a-5p	Inhibiting autophagy, SNHG6 knockdown or overexpression of miR-26a-5p, thus overcoming drug resistance
Colon cancers resistance to KRAS drugs	Two mitochondrial targeted compound 3-Carboxyl proxyl nitroxide (Mito-CP) and Mito-Metformin cause mitochondrial autophagy of KRAS cells by activating AMPK-ULK1 signaling cascade	Overcoming drug resistance of KRAS mutant cells
Non-small cell lung cancer (NSCLC) cells resistance to NK cell-mediated killing	The natural product rocaglamide (RocA) inhibited autophagy by targeting to ULK1 and restored the level of NK cell-derived GZMB (granzyme B) in NSCLC cells increasing their susceptibility to NK cell-mediated killing.	Overcoming resistance of NSCLC cells against NK cell-mediated killing
Breast cancer cell resistance to doxorubicin (DOX), tamoxifen and cisplatin	Mir-489 directly targeting ULK1 and LAPTM4B genes, negatively regulating the expression of ULK1 and LAPTM4B inhibiting autophagy	Overcoming resistance to doxorubicin (DOX), tamoxifen and cisplatin
NSCLC resistance to crizotinib	HOTAIR (HOX transcript antisense intergenic RNA) is a high expression in NSCLC and promotes cisplatin resistance in NSCLC	HOTAIR shRNA transfection overcoming the resistance of A549 cells to crizotinib by inhibiting autophagy activity decreasing the phosphorylation of ULK1
Cisplatin-resistant NSCLC cells	SBI0206965, as a selective inhibitor of ULK1 blocks cisplatin-induced autophagy and promotes cell death	Overcoming NSCLC cells resistance to cisplatin

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
