# Peer review of "A Review of ULK1-Mediated Autophagy in Drug Resistance of Cancer"

_cancers, 2020, doi:10.3390/cancers12020352_

Round 1
Reviewer 1 Report
In this review the authors present a very well organized discussion on the role of ULK 1, a protein kinase, in the process of autophagy and resistance to the action of anticancer drugs. The topic is very interesting; also the bibliographical references are satisfactory as well as the figures and tables included.
it is clear that the authors want to highlight the key role of ulk1 as a possible
target of cancer therapy. Although the available data are not
sufficient, however, the prospects for future studies can be evaluated.
In my opinion the work can be accepted in the present form and the journal Cancer is appropriate for the pubblication.
Author Response
Dear Reviewer,
Thank you for your comments concerning our manuscript. Those comments are all valuable and very helpful for revising and improving our paper. We carefully proofread the manuscript to minimize typographical, grammatical and bibliographical errors and add some new references to further support the topic. We polished the manuscript with a professional assistance in writing, conscientiously. All changes have been marked in the revised manuscript and fully detailed in a response letter. The response letter and certificate of language editing are in the attachment.
Thank you for the kind advice
Sincerely yours,
Li Liu

Reviewer 2 Report
In this review by Liu et al., the authors summarized the effects of ULK1 15 in the regulation of autophagy and the development of drug resistance in cancers. They also discussed its potential as a target for tumor therapy and its impact on the occurrence of drug resistance in tumor cells. The latest research progress was reviewed. This manuscript provided interesting information and may help future clinical application.
Author Response
Dear Reviewer,
Thank you for your commnts concerning our manuscript. Those comments are all valuable and very helpful for revising and improving our paper. We carefully proofread the manuscript to minimize typographical, grammatical, and bibliographical errors and add some new references to further support the topic. We polished the manuscript with a professional assiatance in writing, conscientiously. All changes have been marked in the revised manuscript and fully detailed in a response letter. The response letter and certificate of language editing are in the attachment.
Thank you for the kind advice.
Sincerely yours,
Li Liu

Reviewer 3 Report
The authors have presented a review paper on autophagy in cancer in the context of drug resistance. They have selected ULK1 is one of the signalling regulator of autophagy to describe the role of autophagy in therapeutic resistance.
I have the following concerns about the paper:
English language of the paper needs to be improved. The title should be "A review of ULK1-mediated autophagy in
drug resistance of cancer". There are many major works on ULK1 in autophagy which authors need to mention them. The same thing should be done on autophagy in therapeutic resistance.
Author Response
Dear Reviewer,
Thank you for your comments concerning our manuscript. Those comments are all valuable and very helpful for revising and improving our paper. We have carefully proofread the manuscript to minimize typographical, grammatical, and bibliographical errors and also have re-scrutinized to improve the English language by a professional language polishing service. We have revised the title accordingly. The references on ULK1 in autophagy and autophagy in therapeutic resistance have been supplemented to the manuscript to further suppor the topic. All changes have been marked in the revised manuscript and fully detailed in a response letter. The response letter and certificate of language editing are in the attachment.